# Evolution of Endogenous Retroviruses in the Subfamily of Caprinae

**DOI:** 10.3390/v16030398

**Published:** 2024-03-04

**Authors:** Ali Shoaib Moawad, Fengxu Wang, Yao Zheng, Cai Chen, Ahmed A. Saleh, Jian Hou, Chengyi Song

**Affiliations:** 1College of Animal Science and Technology, Yangzhou University, Yangzhou 225009, China; dh20020@stu.yzu.edu.cn (A.S.M.); mz120180996@yzu.edu.cn (Y.Z.); 007302@yzu.edu.cn (C.C.); elemlak1339@gmail.com (A.A.S.); 2Department of Animal Production, Faculty of Agriculture, Kafrelsheikh University, Kafrelsheikh 33516, Egypt; 3State Key Laboratory of Animal Biotech Breeding, College of Biological Sciences, China Agricultural University, Beijing 100193, China; s20223020273@cau.edu.cn (F.W.); houjian@cau.edu.cn (J.H.); 4Animal and Fish Production Department, Faculty of Agriculture (Alshatby), Alexandria University, Alexandria City 11865, Egypt

**Keywords:** Caprinae, ERVs, genome annotation, gene overlapping, divergence pattern

## Abstract

The interest in endogenous retroviruses (ERVs) has been fueled by their impact on the evolution of the host genome. In this study, we used multiple pipelines to conduct a de novo exploration and annotation of ERVs in 13 species of the Caprinae subfamily. Through analyses of sequence identity, structural organization, and phylogeny, we defined 28 ERV groups within Caprinae, including 19 gamma retrovirus groups and 9 beta retrovirus groups. Notably, we identified four recent and potentially active groups prevalent in the Caprinae genomes. Additionally, our investigation revealed that most long noncoding genes (lncRNA) and protein-coding genes (PC) contain ERV-derived sequences. Specifically, we observed that ERV-derived sequences were present in approximately 75% of protein-coding genes and 81% of lncRNA genes in sheep. Similarly, in goats, ERV-derived sequences were found in approximately 74% of protein-coding genes and 75% of lncRNA genes. Our findings lead to the conclusion that the majority of ERVs in the Caprinae genomes can be categorized as fossils, representing remnants of past retroviral infections that have become permanently integrated into the genomes. Nevertheless, the identification of the Cap_ERV_20, Cap_ERV_21, Cap_ERV_24, and Cap_ERV_25 groups indicates the presence of relatively recent and potentially active ERVs in these genomes. These particular groups may contribute to the ongoing evolution of the Caprinae genome. The identification of putatively active ERVs in the Caprinae genomes raises the possibility of harnessing them for future genetic marker development.

## 1. Introduction

Endogenous retroviruses (ERVs) are a type of genomic element that are found across a wide range of vertebrates, spanning from basal vertebrates like sharks and rays to mammals [1]. Throughout the process of evolution, exogenous retroviruses have integrated into the germ line, leading to the formation of stably integrated endogenous retroviruses. These endogenous retroviruses are subsequently inherited by offspring through vertical transmission. ERVs typically consist of three primary genes: Group-specific antigen (*gag*), which instructs the synthesis of CAPSID proteins; Protease–Polymerase (*pro-pol*), responsible for producing enzymes involved in maturation, replication, Retroviral integrase (*IN*), which inserts the viral DNA into the host chromosomal DNA, and insertion; and Envelope (*env*), where the expression of the env gene enables retroviruses to target and attach to specific cell types and to infiltrate the target cell membrane. These genes are flanked by long terminal repeats (LTRs), which are regulatory regions that contain promoters, enhancers, and polyadenylation signals [1]. The relationship between endogenous and exogenous viral sequences is presumed to be by evolutionary descent, and the presumption is that endogenous viral sequences are derived from exogenous viruses that have infected the host organism’s ancestors in the past and become integrated into their genome through evolutionary processes. ERVs can be classified into three different classes based on their relationships to recognized retroviral genera [2]: Class I ERVs are associated with Epsilon retrovirus and Gamma retrovirus, Class II ERVs with Alpha retrovirus, Beta retrovirus, Delta retrovirus, and Lenti retrovirus, and Class III ERVs with Spuma retrovirus [1].

The interest in these genomic elements has grown significantly due to their potential role in diseases like multiple sclerosis and cancer [3]. Additionally, interest in these genomic elements has been fueled by their impact on the evolution of the host genome, including changes in nearby gene expression, among other factors [4]. In addition, ERVs can domesticate in host genomes and evolve into new functional genes, such as Syncytins (Syncytin-1 and Syncytin-2), which are derived from the Env genes of ERVs, and they play essential roles in placental development [5]. The implications of ERVs’ presence in host genomes have been re-evaluated [6,7]. Since the late 1990s, researchers have investigated the presence of ERVs in certain domestic animals using Southern blot analysis [8] and Polymerase Chain Reaction (PCR) analyses [9]. These studies identified the presence of retroviruses in various domestic animals, including pigs, horses, sheep, goats, cattle, yaks, and cats [8,9]. Nevertheless, the comprehensive investigation of ERVs has typically been focused on humans [10] and murids [11]. In recent years, there has been an increasing trend of analyzing ERVs in a broader range of domestic species.

The Caprinae subfamily is a subset of the Bovidae family of ruminants, primarily consisting of medium-sized bovids. Notable members include sheep, goats, and their relatives. This particular subfamily comprises 12 genera; however, the classification of Caprinae is intricate, resulting in various suggested classifications. To address these ongoing taxonomy issues, the International Union for Conservation of Nature (IUCN) has established a Taxonomy Working Group as part of the Caprinae Specialist Group. Caprinids possess distinct adaptations for montane and alpine environments, which is why this subfamily displays more diversity in Eurasia compared to Africa [12,13,14,15]. Many species have become extinct since the last ice age, predominantly due to human influence. Among the remaining species, five are classified as endangered, eight as vulnerable, and seven as requiring conservation measures due to concerns, but at lower risk [16,17,18,19]. The polymorphisms generated by endogenous Jaagsiekte Sheep Retrovirus (enJSRV) have been used as highly informative genetic markers because the presence of each endogenous retrovirus in the host genome is the result of a single integration event in a single animal and is irreversible, so populations sharing the same provirus in the same genomic location are de facto phylogenetically related [20].

Sheep (*Ovis aries*) have served as valuable models for studying the coevolution between a host and ERVs, specifically the Jaagsiekte Sheep Retrovirus (JSRV) and its endogenous counterpart, enJSRV. The significance of this model lies in the fact that both forms of the retrovirus are actively present and exhibit strong interactions [21]. The remaining ERVs within the sheep genome can be categorized into nine families from Class I (OERV γ1 to γ9) and three families from Class II (OERV β1 to β3) based on the amplification of their pro/pol complex [22]. Given the significant role of enJSRVs in sheep development and placental morphogenesis, and their ability to block the exogenous JSRV, the expression analyses of sheep ERVs have primarily focused on the enJSRV provirus [21].

Recently, the investigation of endogenous retroviruses (ERVs) has extended to both wild and domestic animal populations. However, our understanding of ERV annotation and evolution in livestock genomes remains limited. To address this knowledge gap, our study focuses on the Caprinae subfamily reference genomes, where we utilize multiple pipelines to de novo mine and annotate ERVs. We systematically characterize their classification, coverage, and evolutionary dynamics within these genomes. Furthermore, we investigated the intersection between ERVs and host genes, including protein-encoding and lncRNA genes. Our comprehensive analysis reveals a genome-wide landscape of ERVs across 13 different species within the Caprinae subfamily. This research not only contributes to our understanding of the coevolution between the mobilome and livestock genome but also enhances our comprehension of genome evolution as a whole.

## 2. Materials and Methods

### 2.1. Data Sources of Genomes and ERV Reference Sequences Used in This Study

The reference genomes of 13 species belonging to the Caprinae subfamily were obtained from the National Centre for Biotechnology Information (NCBI) database (https://www.ncbi.nlm.nih.gov, accessed on 14 April 2023) and are listed in Appendix A. To define the classification of Caprinae ERVs, the reference reverse transcriptase (RT) DNA sequences of retroviruses from species other than Caprinae and the previously identified retroviruses in the Caprinae subfamily (containing the reference enJSRVs in sheep) were included (Appendix A), in addition to the reference DNA sequences of major homology region (MHR) of gag. These sequences were downloaded from the NCBI nucleotide database. The GenBank accession numbers of the ERVs used for phylogenetic analysis are listed in Appendix A.

### 2.2. ERVs Mining in the Caprinae Genomes

ERVs were identified from Caprinae genomes using the LTRharvest [23] tool embedded in GenomeTools [24] using the parameters-motif TGCA-minlenltr 100-maxlenltr 7000-similar 90-motifmis 1-mintsd 4-maxtsd 6-overlaps best and RepeatModeler (version 2.0) (http://www.repeatmasker.org/RepeatModeler, accessed on 31 May 2023) using the parameters BuildDatabase-name Genome. LTRharvest is a specialized tool that enables the de novo detection of full-length LTR retrotransposons, including ERVs [23]. To extract the sequences of the genes and their flanking regions from the genomes based on the identified coordinates, the bedtools getfasta tool was employed, bedtools (version 2.27.1) [25]. This tool allows for obtaining sequences from a genome using specified coordinates from a bed file. Sequences that were smaller than 4000 bp or larger than 10,000 bp were excluded from further analysis. Only potential intact sequences within this size range were retained. To obtain full-length copies of the potential intact sequences, about 4 kb upstream and downstream flanking regions were added. This extension was achieved using the slop command (−s, −l 4000, −r 4000) in bedtools.

To further filter and analyze the identified ERVs, only those with potential intact reverse transcriptase (RT) regions were retained. For this purpose, a search for reverse transcriptase sequences was conducted in the sequences detected by LTRharvest and RepeatModeler. The search was performed using the hmmsearch command within the HMMER tool (https://www.ebi.ac.uk/Tools/hmmer/search/hmmscan, accessed on 12 September 2023) [26]. To identify conserved reverse transcriptase sequences, the sequences of well-annotated reverse transcriptase regions in different mammals (specially Caprinae) were downloaded from NCBI and the Pfam [27] and then were converted to Hidden Markov Model (HMM) profiles using the hmmbuild command in HMMER tool. The Pfam database contains Hidden Markov Model (HMM) profiles for protein domains and families. HMM profiles are statistical models that represent multiple sequence alignments and capture conserved motifs and patterns within a protein family. The profiles downloaded from NCBI and the Pfam are specifically designed for reverse transcriptase regions and are used to identify similar sequences in the input data.

To cluster the ERV elements based on sequence similarity and alignment, the Vsearch tool (https://github.com/torognes/vsearch, accessed on 1 October 2023) was utilized. The clustering was performed using the parameters: -id 0.50 to set the similarity threshold to 50%, -clusters to create the clusters, and -msaout to obtain multiple sequence alignments as output. From the clustered sequences, representative sequences were selected for each group based on the integrity of sequences containing potential intact LTRs and the availability of different domains (*gag*, *pro*, *pol*, *IN*, and *env*). This process involved analyzing the relationships between the sequences and selecting a representative sequence that best represents the characteristics of each group.

### 2.3. ERVs Annotation in the Caprinae Genomes

RepeatMasker (version 4.0.9) software [28] was used as a sequence search engine to annotate ERVs in the Caprinae genomes using a custom library, which combined the known repeats in sheep and goat genomes from the Repbase [29]. To identify the functional domains in all predicted ERV groups within the Caprinae subfamily genomes, we translated all six frames of all sequences in each group. This step accounts for the possibility of different reading frames in the sequences. The putatively endogenous retroviruses (ERVs) containing complete retroviral functional domains (*gag*, *pro*, *pol*, *IN*, and *env*) were translated by Genscan (http://hollywood.mit.edu/GENSCAN.html, accessed on 11 October 2023). The translated sequences were then searched against the Pfam database using the hmmsearch command [30]. By comparing the translated sequences to the Pfam database, we were able to identify the functional domains present in the predicted ERV groups. To search for retroviral genes in the Caprinae subfamily genomes, we chose to utilize HMMER. We obtained HMM profiles specific to retroviral genera such as Gamma retrovirus, Epsilon retrovirus, Alpha retrovirus, Beta retrovirus, Delta retrovirus, Lenti retrovirus, Spuma retrovirus, and Gypsy virus. Additionally, we retrieved the sequences for specific genes or regions within retroviruses, including *gag*, *pro*, *pol*, *IN*, and *env*. These sequences were obtained from the NCBI Database (https://www.ncbi.nlm.nih.gov, accessed on 15 October 2023) and converted to HMM profiles using the hmmbuild command in the HMMER tool.

### 2.4. Phylogenetic Analysis

We generated phylogenetic trees based on alignments of two regions of the genome: the DNA sequences of reverse transcriptase (RT) of *pol* and the major homology region (MHR) of *gag*. To construct multiple alignments of these regions from ERV retrotransposons, we utilized the MAFFT program v.7.310 [31]. We opted to use DNA sequences instead of amino acid sequences for constructing multiple alignments and building the phylogenetic trees due to the presence of frame-shift mutations and stop codons in the ancient retrotransposon elements. These mutations can disrupt the reading frame and introduce premature stops, rendering the translation of these sequences into amino acids unreliable. To build the phylogenetic tree, we used the IQ-TREE program [32] and the parameters-m TEST-bb 1000-nt AUTO, which is a software tool specifically designed for phylogenetic analysis.

### 2.5. Divergence Analysis of Full ERVs

To identify any “active” elements of ERVs, each, of 13, reference genome of Caprinae species was masked using RepeatMasker (version 4.0.9, -nolow) [28] with the custom repeat library of the full-length ERV elements generated in Section 2.3. Then, the Kimura two-parameter distance (divergence) was determined using RepeatMasker’s calcDivergenceFromAlign.pl package from the RepeatMasker software [28].

### 2.6. PCR Verification

To identify the potential retrotransposon insertion polymorphisms (RIPs) in Caprinae, we mapped the upstream and downstream flanking sequences (500 bp) in the four full-length ERV groups to the reference genome using Blat [33]. The differential insertions, designated as putative ERV insertion polymorphisms, were obtained using a bedtools window (-w 50, -v) [25]. The ERV insertions that did not fall into the same window (ERV insertion site and 400 bp flanking region) were considered to be putative ERV RIPs.

Six breeds of sheep (Barki, Rahmani, Rahmani *x* Barki crossbred, Awassi, Ossimi, and Sulffok) were used for PCR verification of full-length ERV polymorphisms. From each sheep breed, five individual DNA samples were isolated from blood using a DNA isolation-kit (Tiangen Biotech, Beijing, China) and then pooled together and subjected to PCR analysis in order to detect the predicted ERV insertion polymorphisms. Considering that most full-length ERVs have a length exceeding 8 kb, double PCR primer pairs were designed and utilized for genotype verification. The strategy employed for primer design is illustrated in Figure 1. The LTR and side primers for ERV-RIPs were designed using Oligo v.7 software and are presented in Appendix A. The primers were synthesized by Vazyme Biotech. Co., Ltd., Nanjing, China. Amplification was performed using Green-SuperMix from TaKaRa, Japan, with 10 pM of primers and 100 ng of genomic DNA per sample. The PCR consisted of 34 cycles, including denaturation at 95 °C for 1 min, annealing at 58–60 °C for 1 min, extension at 72 °C for 1 min, and a final extension step at 72 °C for 2 min. The amplification process was carried out using T100 Thermal Cycler (BIO RAD, Singapore).

### 2.7. Genotyping Analysis

We employed double PCR primer pairs (LTR_Primer and Side_Primer) for each potential ERV-RIP sequence. After PCR amplification, if the LTR primer showed a band between 250 and 750 bp but the Side_Primer, which produces a band within a size ranging between 1000 and 1500 bp, was absent, it indicated homozygous ERV insertion and was labeled as ERV+/+. On the contrary, if the longer band within the size range 1000–1500 bp appeared, and no short band (250–750 bp) was present, it suggested an absence of homozygous ERV insertion genotype (labeled as ERV−/−). If both shorter and longer bands were observed, it signified that this RIP genotype must be heterozygous (labeled as ERV+/−). The methodology employed in this part of the study was based on the framework proposed by Du et al. [34].

### 2.8. Intersection Analysis

Using the intersect program in bedtools (version 2.27.1) [25], we analyzed the distribution bias of retrotransposons in sheep and goat genomes and their intersections with host genes, including protein-encoding genes and lncRNA genes. The bedtools intersect makes it possible to examine overlaps between two sets of genomic features. Moreover, it allows one to have fine control as to how the intersections are reported. The genes were obtained from the annotation files of sheep and goats, downloaded from the NCBI database (https://www.ncbi.nlm.nih.gov, accessed on 19 November 2023), and intersected with the results of the RepeatMasker (version 4.0.9) software [28]. RepeatMasker was used to annotate all ERVs in the sheep and goat genomes using a custom library of the 28 ERV elements identified in Section 2.2.

## 3. Results

### 3.1. ERV Mining in the Genomes of Caprinae

ERV-derived elements were extracted from 13 genomes of the Caprinae subfamily using the LTRharvest and RepeatModeler pipelines. The number of ERVs obtained varied across species, with LTRharvest outputs between 79,529 in *Capra aegagrus* and 215,180 in *Ovis aries*, and RepeatModeler ranging between 118,052 in *Ovis aries* and 211,456 in *Capra hircus* (Appendix A). ERV-derived elements that lacked the reverse transcriptase (RT) regions were then excluded. ERVs with potential intact RT regions ranged from 2503 in *Capra ibex* to 6760 in *Capra hircus*. When considering the number of full-length ERV copies, including *gag*, *pro*, *pol*, *IN*, and *env* domains + LTRs, variations were observed among the different genera of the Caprinae subfamily. A total of 218, 131, 71, 119, and 106 full-length ERVs were detected in the species *O. aries*, *Ovis orientalis*, *Ovis ammon*, *Ovis canadensis*, and *Ovis nivicola* of the *Ovis* genus, respectively. In the Capra genus, 226, 96, 57, and 100 full-length ERVs were identified in *Capra hircus*, *Capra aegagrus*, *Capra ibex*, and *Capra sibirica*, respectively. The genera of relatives yielded 97 full-length ERVs in *Hemitragus hylocrius*, 65 in *Budorcas taxicolor*, 70 in *Oreamnos americanus*, and 126 in *Ovibos moschatus* (Appendix A).

### 3.2. Classification of ERVs in the Genomes of Caprinae

The number of ERVs containing reverse transcriptase (RT) regions varied significantly across different types of retroviruses and groups. By utilizing the sequences and constructing a phylogenetic tree, we were able to accurately assess the evolutionary relationships and genetic distances among the ERV retrotransposon elements in the Caprinae genomes. Through phylogenetic analysis, we identified 28 groups of Caprinae ERVs (Cap_ERV_1-28), 22 new groups, and 6 previously reported ERV groups (Table 1). Gamma retroviruses generally had a higher number of ERV elements compared to beta retroviruses. Specifically, 19 groups were classified as Class 1 gamma retroviruses, while 9 groups were classified as Class 2 beta retroviruses, together with several reference sequences of enJSRVs in sheep (Figure 2A). The same classification was detected using the major homology region (MHR) of gag from the representative sequences of groups (Figure 2B). The different genes and their domains located in 28 groups of Caprinae ERVs (Cap_ERV_1-28) are presented in Appendix A.

The lengths of the Cap_ERV elements ranged from 4421 to 9580 bp, with the length of the long terminal repeat (LTR) varying from 143 to 583 bp (Table 1). To identify the functional domains within each Cap_ERV element, we translated all sequences in each group through all six frames and searched against the Pfam database using hmmsearch [30]. We successfully identified multiple retrovirally relevant domains. ERV candidates featuring two LTRs and structural polyproteins common to all retroviruses, namely group-specific antigen (*gag*), protease (*Pro*), polymerase (*pol*), integrase (*IN*), and envelope (*env*), were classified as full-length ERVs. Most of the ERV groups in the Caprinae subfamily had undergone decay and were presumed to be inactive.

Only four groups (Cap_ERV_20, Cap_ERV_21, Cap_ERV_24, and Cap_ERV_25) of ERV candidates, classified as beta retroviruses, were found to process full-length ERVs, with all retroviral functional proteins (*gag*, *pro*, *pol*, *IN*, and *env*) being detectable (Table 1). They may potentially retain the functionality of ERVs and, thus, were designated as putatively active ERVs. A total of 1482 full-length copies of potentially active ERVs were identified in 13 species of Caprinae subfamily (Appendix A), distributed in groups as follows: 418 in Cap_ERV_20, 331 in Cap_ERV_21, 479 in Cap_ERV_24, and 254 in Cap_ERV_25 (Table 2). Moreover, a primer binding site (PBSLys, PBSIle, PBSLys, and PBSHis) of 18 nucleotides downstream and adjacent to the 5′-LTR was identified in the Cap_ERV_20, Cap_ERV_21, Cap_ERV_24, and Cap_ERV_25, respectively. Additionally, a small region of polypurine tract (PPT) was found upstream and adjacent to the 3′-LTR. The structure organization of putatively active ERVs detected in the four full-length ERV elements is summarized in Appendix A.

### 3.3. Distribution and Genomic Coverages of ERVs in Different Species of Caprinae

In this study, the analysis using LTRharvest and RepeatModeler identified a total of 28 Cap_ERV groups present in each one of all species analyzed, indicating their presence across all members of the Caprinae subfamily (Figure 3A). Among these groups, 19 were classified as Class I ERVs, while the remaining 9 were classified as Class II ERVs. This suggests that the Cap_ERV elements were inserted into the genome of the common ancestor of the Caprinae subfamily around 11 million years ago (MYA) or possibly even earlier. The abundance of detected copies for each Cap_ERV group varied, and this information is summarized in Appendix A. Among all the groups, the ones with the highest abundance, determined by the number of matches or mapped reads, were Cap_ERV_7 in *Ovis aries* (sheep), Cap_ERV_18 in *Capra hircus* (goat), Cap_ERV_13 in *Budorcas taxicolor*, and Cap_ERV_19 in the remaining species. The analysis of a single reference genome for each of the 13 species in the Caprinae subfamily reveals variation in the numbers of endogenous retroviruses (ERVs) annotated by RepeaMasker (Figure 3B and Appendix A). Furthermore, there is variability in the proportion of endogenous retroviruses (ERVs) about the total amount of retrotransposons. The *Budorcas taxicolor* genome has the highest proportion of ERVs at 10.63%. In contrast, *Capra ibex* has the lowest proportion of ERVs in its genome, with a value of 6.89% (Figure 3C).

### 3.4. Kimura Divergence and Genotyping Analysis of Recent and Potentially Active ERVs in the Genomes of Caprinae

The analysis of ERV divergence provides insights into the evolutionary history of ERVs. Our findings indicate that out of all the ERV candidates analyzed, only four were identified as full-length ERVs. These four ERVs belong to the Cap_ERV_20, Cap_ERV_21, Cap_ERV_24, and Cap_ERV_25 groups of beta retroviruses (Figure 2A and Table 1). Kimura (K) divergence analysis revealed that the predominant recent activities of the four full-length ERV elements were observed for all 13 genomes of Caprinae (Appendix A), with some ERV copies of these elements representing K divergence less than 5%. However, a few copies of putatively active ERVs display recent activity and represent less than 2% of K divergence, suggesting they are recent invaders of the genome, and may potentially retain functionality.

Four insertion polymorphisms in each group of full-length ERVs in the reference genome of sheep were evaluated by PCR using two pairs of primers (LTR_Primer and Side_Primer) with genomic DNA samples from six sheep breeds, as described in the methods, for proving the activity, and the genotyping results are shown in Figure 4. From the electrophoretogram results, we found that 15, 12, 16, and 6 of ERV genotypes in sheep were ERV+/−, while ERV+/+ represented 8, 12, 3, and 7 genotypes in Cap_ERV_20, Cap_ERV_21, Cap_ERV_24, and Cap_ERV_25, respectively. Regarding the ERV−/− genotype, it was observed only in Cap_ERV_25 (10 genotypes); the different genotypes are presented in Appendix A. Ultimately, the four full-length ERV elements were confirmed by PCR, indicating that full-length ERV insertions are highly polymorphic, as they were genotyped in a small population (30 individuals) of studied sheep breeds, and all showed polymorphism (as shown in Figure 4 and Appendix A).

### 3.5. Most lncRNAs and Protein-Coding Genes Contain ERV-Derived Sequences in Sheep and Goats

The intersection analysis revealed that most lncRNAs and protein-coding genes contain ERV-derived sequences in sheep and goats. Specifically, approximately 75.26% (15,999) of protein-coding genes and 80.66% (3583) of lncRNA genes in sheep contained ERV-derived sequences. Similarly, in goats, 73.64% (15,150) of protein-coding genes and 75.47% (1471) of lncRNA genes contained ERV-derived sequences (Figure 5A). On the other hand, in sheep, it was found that 24.22% (743,666) of ERV insertions (3,008,458) overlapped with protein-coding genes, while 1.63% (48,955) overlapped with lncRNA genes. Furthermore, 0.186% (5607) of ERV insertions were found to overlap with both protein-coding and lncRNA genes. Similarly, in goats, 22.6% (691,235) of ERV insertions (3,057,947) overlapped with protein-coding genes, and 0.75% (22,957) overlapped with lncRNA genes (Figure 5B). These findings indicate that ERV-derived sequences contribute to the evolution of both lncRNA and protein-coding genes. In fact, ERVs themselves may be lncRNAs through transcription from their 3’LTR.

Concerning of the ERV coverages of various genic features, it revealed noticeable biases in their ERV composition. In sheep, a significant depletion of ERVs was observed in transcribed regions, including PCgene, mRNA, transcript, pseudogene, and exons, with a sequence coverage of approximately 5% or lower. However, no significant bias was observed in the distribution of ERVs in the lnc_RNA gene (19.36%), cDNA_match (53.15%), coding sequence CDS (29.10%), and tRNA (37.68%). In goats, the ERV coverages varied for different genic features, with percentages as follows: PCgene (17.08%), lnc_RNAgene (19.92%), mRNA (8.85%), transcript (13.25%), pseudogene (26.55%), exons (32.72%), cDNA_match (21.30%), coding sequence CDS (40.05%), V_gene_segment (28.81%), C_gene_segment (16.19%), and tRNA (35.18%), as shown in Appendix A.

## 4. Discussion

The mammalian genomes are mainly composed of three major classes of retrotransposons: long interspersed nuclear elements (LINEs), short interspersed nuclear elements (SINEs), and LTR-retrotransposons. ERVs are the major retrotransposons within mammalian genomes, accounting for 5–10% of the genomic sequences [35,36,37,38]. In this study, we conducted a comprehensive analysis to identify and characterize ERV groups in 13 single reference genomes belonging to 13 different species of the Caprinae subfamily, noting variations in their domain fractions among different species. To the best of our knowledge, this is the first study to provide genome-wide predictions of ERVs at the domain level across a wide range of Caprinae species. ERVs, being the primary long terminal repeat (LTR) retrotransposons found in mammalian genomes, have been integrated into mammalian lineages for over 100 million years [39]. Here, we observed that ERVs exhibit varying genomic coverages within the Caprinae subfamily. The range of genomic coverage for ERVs in the Caprinae subfamily was found to be between 6.89% in the genome of *Capra ibex* and 10.63% in the genome of *Budorcas taxicolor*. These findings are consistent with what has been observed in the genomes of other mammalian species, where ERVs also tend to occupy a substantial proportion of the genome [36,40,41,42,43].

Within the species examined in this study, we identified thousands of ERV candidates. It was determined that the majority of these ERVs had undergone decay. However, among these candidates, a subset was found to contain intact reverse transcriptase (RT) regions. The number of ERV candidates with intact RT regions varied from 2503 in *Capra ibex* to 6760 in *Capra hircus*. Through phylogenetic analysis, these RT-motif-containing candidates were classified into 19 gamma ERVs and 9 beta ERVs similar to the reference enJSRVs in sheep. The same classification of ERV groups was detected using the major homology region (MHR). The gag gene is poorly conserved among ERVs, but all detected Cap_ERV groups contain an MHR. The retroviral Gag protein exhibits extensive amino acid sequence variation overall; however, one region of Gag, termed the major homology region (MHR), is conserved among all retroviruses [44], which is consistent with our results. Based on divergence analysis of four full-length ERVs among the 13 species of Caprinae (Appendix A), some copies of these elements exhibited an extended period of expansion and indicated increased activity in recent years. It is noteworthy that a few copies of these groups encoded long peptides with intact *gag*, *pro*, *pol*, *IN*, and *env* domains. These findings align with similar observations made in the pig genome, further supporting our results [40,45]. Based on our findings, it can be concluded that the majority of ERVs in the genomes of Caprinae are considered to be fossils, meaning they are remnants of ancient retroviral infections that have become permanently integrated into the genomes. However, the presence of the Cap_ERV_20, Cap_ERV_21, Cap_ERV_24, and Cap_ERV_25 elements suggest the existence of potentially active ERVs. This may play a role in the ongoing evolution of the Caprinae genome [46,47].

The presence of full-length ERV insertion polymorphisms, which typically consist of large structural variations exceeding 5 kb, can result in a stronger genetic impact and can be considered as an indicator of activity [34]. In this study, we performed PCR verification and genotyping analysis on six sheep breeds, with five individuals from each breed, using the primers of four RIPs from each group of full-length ERVs. We obtained insertion polymorphic sites. Subsequently, we confirmed that these insertions belong to the four full-length ERVs. This finding indicates that certain full-length ERVs still retain the capability to mobilize and influence the sheep genomes, some of which may contribute to phenotypic variations and population differentiation, despite the majority of ERVs in sheep genomes being considered as fossils.

Early studies on genome sequencing have revealed variations in the activity of retroviruses across mammalian species. Mice contain numerous active ERV elements [41], while several domesticated animals, including pigs [40], rabbits [48], and domestic cats [49], reported previously, and goat and sheep reported here, were found to primarily possess inactive ERV elements, but few of these copies are putative active. These differences in retroviral activity among species contribute to our understanding of the evolutionary dynamics and potential impacts of ERVs in different genomes of mammals [36]. We observed an extensive intersection of ERV-derived sequences with genes in the genomes of sheep and goats. In sheep, ERV-derived sequences were found in approximately 75% of protein-coding genes and 81% of lncRNA genes. Similarly, in goats, ERV-derived sequences were present in about 74% of protein-coding genes and 75% of lncRNA genes, indicating that ERVs substantially contribute to the evolution of both genes and genomes in sheep and goats, exemplifying a characteristic feature of mammals [36,41,50].

## 5. Conclusions

In the present study, de novo mining of ERVs in 13 species of the Caprinae subfamily was performed, allowing us to define their diversity at a group level and the structural organization, evolutionary dynamics, and distribution within the genome. ERVs were classified into 28 groups (Cap_ERV_1–28) in terms of sequence identity, structural organization, and the phylogenetic analysis of the sequences. Differential evolutionary dynamics across these groups were recorded. Most of the ERVs had decayed, and only Cap_ERV_20, Cap_ERV_21, Cap_ERV_24, and Cap_ERV_25 showed signs of activity in recent years, with a few copies encoding long peptides with intact *gag*, *pro*, *pol*, *IN*, and *env* domains. We also investigated the intersection of ERV insertions with lncRNA and protein-coding genes, and we found that the majority of lncRNA and protein-coding genes contain ERV-derived sequences in both sheep and goats. These findings describe ERVs’ evolution in the Caprinae subfamily and provide a better understanding of genomic evolution in mammals. Furthermore, putatively active ERVs may allow future genetic marker development, which could have considerable potential for applications in quantitative trait locus (QTL) mapping and molecular breeding in the Caprinae, and so are worthy of additional evaluation.

## Figures and Tables

**Figure 1 viruses-16-00398-f001:**
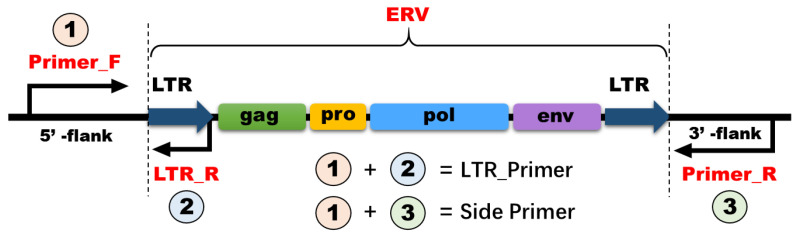
Design principles for full-length endogenous retrovirus (ERV) polymorphic primers. (1 & 2) LTR_Primer produces a shorter band ranging between 250 and 750 bp. (1 & 3) Side_Primer produces a band within a size ranging between 1000 and 1500 bp.

**Figure 2 viruses-16-00398-f002:**
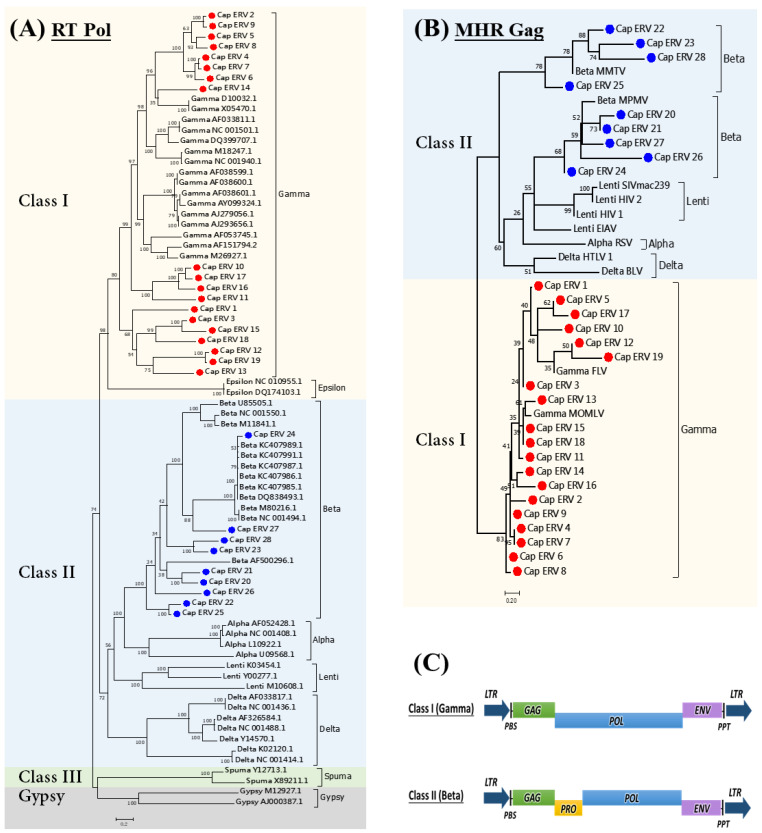
Classification of endogenous retroviruses (ERVs) in the genomes of Caprinae; (**A**) a phylogenetic tree was constructed using representative retroviral DNA sequences, aligned with MAFFT and analyzed with the IQ-TREE program. Reference reverse transcriptase (RT) sequences from various mammalian species were included for comparison, including reference RT sequences of enJSRV (DQ838493.1, KC407989.1, KC407987.1, KC407986.1, KC407985.1, and KC407991.1). (**B**) Another phylogenetic tree was generated using MAFFT alignment and IQ-TREE program, focusing on the reference major homology region (MHR) of gag sequences for comparison. The ERVs were categorized into 28 ERV groups (Cap_ERV_1–28, represented by colored dots). (**C**) Two classes of ERVs, Class I (gamma retroviruses) and Class II (beta retroviruses), in the Caprinae subfamily are depicted in structural schematics for clarification.

**Figure 3 viruses-16-00398-f003:**
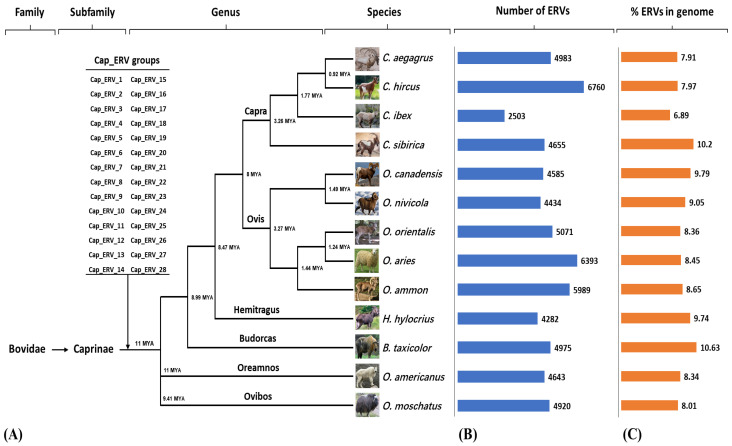
Landscape of Cap_ERV elements in Caprinae subfamily; (**A**) the distribution of Cap_ERV groups in the ancestral lineage of the Caprinae subfamily, encompassing six genera and various species. The divergence times are determined using Timetree (http://www.timetree.org, accessed on 22 October 2023) and are represented in million years ago (MYA). (**B**) The number of ERVs identified in thirteen species within the Caprinae subfamily through the utilization of LTRharvest and RepeatModeler tools. (**C**) The relative proportions of ERVs, calculated as the size of ERVs relative to the total genome size, expressed as a percentage (ERV size/Total genome size × 100), identified within the genomes of diverse species belonging to the Caprinae subfamily.

**Figure 4 viruses-16-00398-f004:**
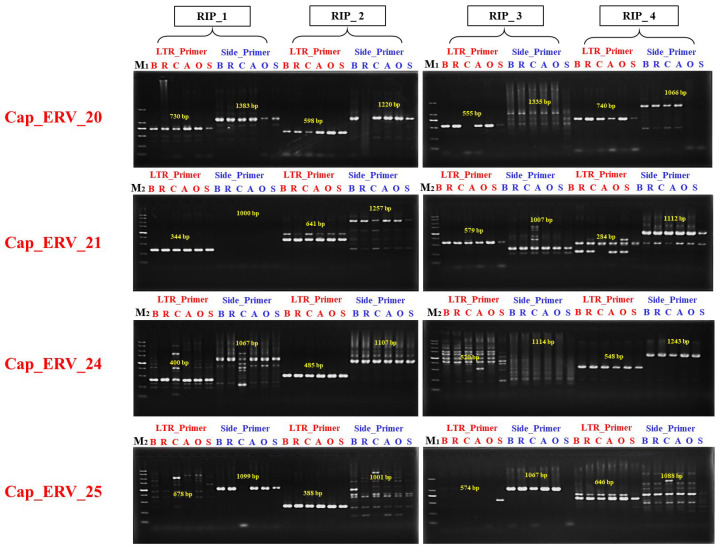
Polymorphism identification of selected full-length ERV elements; Cap_ERV_20, 21, 24, and 25 in 6 sheep breeds; Barki (B), Rahmani (R), Rahmani *x* Barki crossbred (C), Awassi (A), Ossimi (O), and Sulffok (S) using Polymerase Chain Reaction (PCR) approach for blood samples of studied sheep breeds. M1: represents DNA marker (2000 bp) while M2: represents DNA marker (5000 bp). The target product length of the bands is written in yellow numbers.

**Figure 5 viruses-16-00398-f005:**
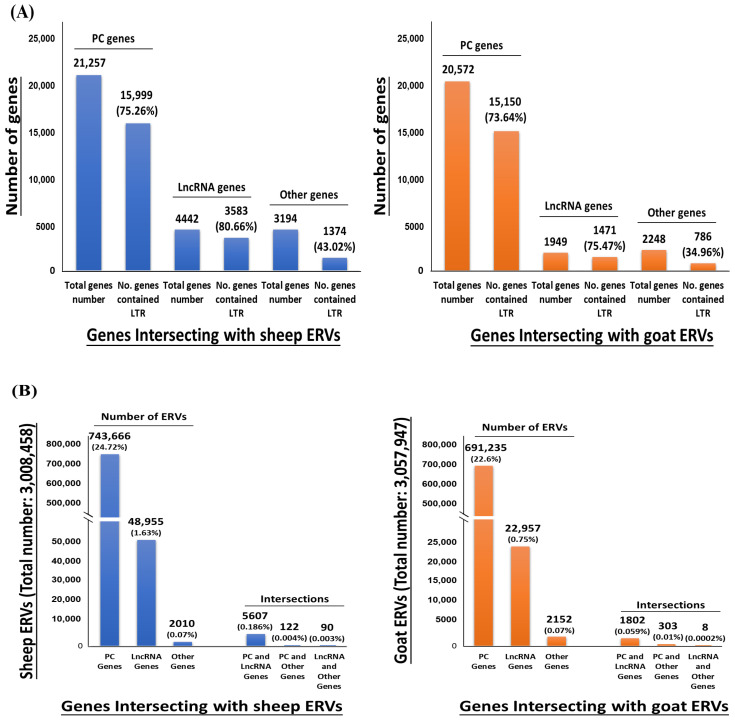
Contribution of ERV elements to protein coding and lncRNA genes in both sheep and goats. (**A**) The proportion of genes intersecting with ERVs in each protein-coding gene (PC gene), and long noncoding gene (lncRNA gene). (**B**) The proportion of ERVs intersecting with PC and lncRNA genes.

**Table 1 viruses-16-00398-t001:** Detailed information of obtained Cap_ERV elements in Caprinae genomes.

Groups	Repbase Name	No of ERVs	Length						
			Representative Sequence (bp)	LTR (bp)	*gag* (aa)	*pro* (aa)	*pol* (aa)	*IN* (aa)	*env* (aa)
Class I (Gamma retroviruses)									
Cap_ERV_1	OviAri-5.324_LTR	1425	8749	260	301	-	891	-	156
Cap_ERV_2	-	1123	8668	440	364	-	1021	-	616
Cap_ERV_3	-	2872	8530	292	443	-	932	-	471
Cap_ERV_4	OviAri-5.2557_int	1759	8123	264	559	-	1176	-	629
Cap_ERV_5	-	1655	8294	388	341	-	1010	-	385
Cap_ERV_6	-	946	7492	257	137	-	451	57	-
Cap_ERV_7	-	1792	7316	274	275	-	514	-	290
Cap_ERV_8	-	525	7049	388	488	-	633	-	302
Cap_ERV_9	-	970	6894	436	526	-	569	-	301
Cap_ERV_10	OviAri-1.272_LTR	2469	7428	381	126	-	333	-	98
Cap_ERV_11	-	6219	7316	149	179	-	286	-	103
Cap_ERV_12	OviAri-6.2056	3190	7762	369	102	-	452	37	-
Cap_ERV_13	OviAri-1.306	6750	8198	442	565	-	1168	-	468
Cap_ERV_14	OviAri-3.284_LTR	741	6889	412	241	-	523	57	321
Cap_ERV_15	-	757	6754	276	553	-	376	-	539
Cap_ERV_16	-	924	7459	380	348	-	360	-	145
Cap_ERV_17	-	1453	5920	156	171	-	306	-	111
Cap_ERV_18	-	3986	6687	183	168	-	349	-	113
Cap_ERV_19	-	7277	9580	374	257	-	450	-	103
Class II (Beta retroviruses)									
Cap_ERV_20	-	1985	7745	303	532	235	702	44	159
Cap_ERV_21	-	1470	7297	583	527	236	572	44	338
Cap_ERV_22	-	274	8844	250	62	182	342	-	-
Cap_ERV_23	-	5472	7651	158	54	105	58	-	-
Cap_ERV_24	-	1403	7870	410	616	289	826	46	618
Cap_ERV_25	-	472	7691	437	524	199	393	43	160
Cap_ERV_26	-	439	6435	143	88	217	508	43	-
Cap_ERV_27	-	3294	8439	265	443	94	661	42	-
Cap_ERV_28	-	2551	4421	330	125	127	401	-	-

Note: ERVs: Endogenous retroviruses, LTR: Long terminal repeats, *gag*: Group-specific antigen, *pro*: Protease, *pol*: Polymerase, *IN*: Integrase, *env*: Envelope protein, bp: base pair, and aa: amino acid.

**Table 2 viruses-16-00398-t002:** Number of recent and potentially active full-length ERVs detected in Caprinae genomes.

Species		Recent and Potentially Active Full-Length Copies of ERVs			
		Cap_ERV_20	Cap_ERV_21	Cap_ERV_24	Cap_ERV_25
Sheep	*Ovis aries*	60	52	65	41
	*Ovis ammon*	15	18	21	17
	*Ovis canadensis*	34	31	33	21
	*Ovis nivicola*	33	26	28	19
	*Ovis orientalis*	40	32	37	22
Goats	*Capra hircus*	49	34	116	27
	*Capra aegagrus*	34	17	27	18
	*Capra ibex*	18	14	20	5
	*Capra sibirica*	28	21	34	17
Relatives	*Hemitragus hylocrius*	28	18	36	15
	*Budorcas taxicolor*	18	21	14	12
	*Oreamnos americanus*	25	18	16	11
	*Ovibos moschatus*	36	29	32	29
Total		418	331	479	254

## Data Availability

All data needed to evaluate the conclusions in this paper are presented either in the main text or the Appendix A.

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
