# Peer review of "Evolution of Endogenous Retroviruses in the Subfamily of Caprinae"

_viruses, 2024, doi:10.3390/v16030398_

Round 1
Reviewer 1 Report
Comments and Suggestions for Authors
The manuscript by Ali Shoaib Moawad et al. concerns an analysis of full-length endogenous retroviruses in 13 caprine genomes. The submission gives a good overview of a not yet exhaustively researched subject, and is generally well-written and relatively easy to follow. However, there are still many parts that need clarification, while some textbook-like paragraphs can be omitted, and conclusions should be modified. Especially the notion that recent, full-length ERVs have been detected, is not true, as more detailed analysis is needed for that. All that was shown is that relatively little diverged ERVs, containing domains related to all major retroviral ORFs, are present in caprine genomes. Also, I cannot find in Fig. 1 or elsewhere, any novel, or reference, ERVs with homology to enJRSV? Did you not detect them, or are those ERVs not complete, or? Please comment
Specific remarks:
Lines 158-161, and other places: be careful with assigning intactness and functionality to ERVs which have only been characterized at a superficial level. Mutations and indels may be present anywhere in the genome
Section 2.4: elaborate on the settings for the NJ analysis
Line 181: Why is translation for RT domains unreliable, but not for gag? As shown in Fig. 1B
Lines 185-187 should be part of the results or discussion, not the methods
Lines 193/section 2.5: which Kimura-divergence is meant? The K2P model? There are many other models, also several by Kimura. Do not explain what evolutionary divergence is (thus, delete lines 193-200), your readers will know this. Why a certain model was chosen can be explained, though
Line 220: for many retroviruses, such as gamma, lenti etc., pro is not encoded in a separate reading frame from pol, but is there, at the same genomic location
Line 222: full-length, not full
Lines 226-227: remove parentheses around numbers
Fig. 1C: a pro ORF is also present in the gamma retroviruses, only it is in frame with pol, in contrast to the situation in the beta-retroviruses
Line 249: a max of 583 bp is very short for betaretroviral LTRs. Please comment
Line 253: integrase is also a structural protein for retroviruses when pro is included as one. Since in some retroviruses, see above, the pro ORF is in the same frame as is the rest of pol
Lines 271-272: general information can be deleted, but please give the specific PBS, with the tRNA denomination, detected for the retrieved full-length proviruses, e.g. PBSlys1,2, PBSlys3, PBSpro etc.
Lines 273-276: “was responsible….” can be deleted, as this is generally known, and was not shown in the present study
Table 2: Why not mention integrase if pro is listed? Also, reconsider the use of “modern” as this is not a general term, and the completeness or capability to transpose has not been shown, not the divergence between the LTRs has been calculated as a measure for a recent origin. Do all the listed ERVs belong to the Cap_ERV_21 family? "Completed" should be "complete"
Lines 300-303: It should be mentioned hat single genomes have been analyzed per species. If ERVs are young, heterozygous integrations may be present, so numbers are always debatable
Lines 324-325: Kimura distances are not percentages! And there is no cut-off value for ‘recent’ integrations other than zero, or very little, differences between the LTRs
Line 335: with “overlapped with protein-coding genes”, ERVs integrated in introns are meant? And would ERVs not be themselves lncRNA genes? How can they overlap with lncRNA genes, as these are relatively short? Antisense transcription from the 3’ LTR has been demonstrated for many retroviruses. Check the statements that ERVS overlap with lncRNAs throughout the manuscript, including the abstract, as this is a remarkable finding, which likely shows that ERVs are lncRNAs
Line 353: what are V_gene_segment and C_gene_segements? As it stands now, the summary is useless, and can be deleted. Same for Table 3. Or move to supplementary info
Line 367/373: LTRs are no retrotransposons. What is meant here are LTR-retrotransposons, as, in contrast, SINEs and LINEs do not have LTRs. Delete lines 369-375, do not elaborate in a discussion on LINEs etc, as you did not research them
Lines 376-387: add a note on using only single genomes per species for analysis. Heterozygosity!
Lines 390-411: the discussion on retrieving complete, or full-length ERVs should be modified, as all that was shown is that there are ERV integrations with intact domains, not even intact ORFs. Check throughout the manuscript
Lines 408-410 should be reworded, as this has not been shown at all. Much more detailed information is needed to state whether or not the integrations are recent, intact, coding, or have retained the capacity to be expressed or even be infectious
Line 419: ref [38] is about sequencing the human genome?
Line 421: No, they are not commonly referred as ‘modern’ ERVs
Line 424: no cap on rabbit
Lines 435-436: how do you know regions were intolerant of insertions? Are there solitary LTRs present to show most of the ERV genome was removed? Or any other evidence?
Lines 436-441 can be deleted, as most of it has not been shown, or is common knowledge
Line 447: not ‘creation of a phylogenetic tree” should be the aim, but phylogenetic analysis of the sequences
Line 452: did the analysis really show this preference of integration?
Pay attention to the author abbreviations in Refs 2, 21, 33
Comments on the Quality of English LanguageThe English grammar is geenrally fine, some minor modifications will suffice
Reviewer 2 Report
Comments and Suggestions for Authors
The manuscript by Moawad and coauthors provides a survey of ERV elements in the genome assemblies of 13 Caprinae species, reporting 28 ERV groups of which 19 were classified as Class I gamma retroviruses and 9 as Class II beta retroviruses. Interestingly, authors identified some “modern” ERV elements that can retain protein coding activity.
Overall, the work is interesting and scientific sound, but some points needs to be addressed prior to publication:
- The trees presented in Figure 1 have no bootstrap values, and this do not allow to properly evaluate the phylogenetic relations among the different ERVs and their statistical support. Tree should be generated again, and the obtained phylogenies should be tested by bootstrap calculation.
- Is there any evidence of accessory genes in the identified Beta ERVs, as reported for example for different HML groups in primates?
- In general, I do not agree with the use of the term “family” to define the individua l HERV groups, since in retroviral taxonomy it indicates the whole Retroviridae family, and hence this term as referred to HERV groups is misleading.
- The results of paragraph 3.5 are overstated when authors conclude that “These findings indicate a significant impact of ERVs on both lncRNA and protein-coding genes” as based on the sole fact that a proportion of them overlap with genes coordinates. The mere colocalization is in fact not sufficient to conclude any functional impact and does not imply that these ERV sequences are localized in exons and included in the derived transcripts. Authors should perform additional analysis to verify such hypothesis, e.g through transcriptome analysis for the presence of chimeric ERV-gene transcripts, or tone down their conclusions here and in the corresponding discussion and conclusion sections.
- As a suggestion for further studies, given the signs of a possible recent activity for a subset of the identify ERVs, I would verify in individual Caprinae genomes the occurrence of unfixed polymorphic integrations that could not be represented in the reference genome assemblies, as done for the human HML2 group (see for instance 10.1128/JVI.00919-14)
Minor points:
- line 44: remove protein from the Envelope gene name and briefly explain the env functional role, as done for the previous two genes. In addition, gene names should be in italics and protein names should be capitalized: please review this throughout the text (eg. Line 44, line 254, Table 1, line 263, line 266, etc.)
- I would briefly mention in the introduction the physiological roles of syncytins, as an example of crucial contribution of ERVs to the host development, in a process of convergent evolution across eutherian mammals. Do the Authors found evidences of active Env genes? This could be an interesting focus for its possible biological significance on the host.
- Line 230: the abbreviation for endogenous retroviruses has already been introduced
Comments on the Quality of English LanguageEnglish looks fine, the main points regards the use of some nomenclatures and abbreviations, as reported in the previous section.
Round 2
Reviewer 1 Report
Comments and Suggestions for Authors
The revised version of the manuscript by Moawad et al has been substantially improved, especially since the authors have performed additional analyses, such as integration site amplification by PCR in animal samples, and have toned down the implications of their research. Unfortunately, no track changes or highlights were provided with the revised version, so it is not easy to check what exactly has been changed, but overall it reads much better now.
Some minor modifications:
line 163: maybe integrase should be mentioned somewhere in the main text, not only in the suppl. info
lines 250-252: 'full-length', not 'full'
Table 2: "active", or remove. 'Active' is still not a good description of those ERVs, but I agree that a descriptive name should be given to potentially full-length, or possible active ERVs. The description in the abstract, "relatively recent and potentially active ERVs" would suit better, but is of course very long.
line 364: 'contain', not 'containing'
Section 3.5: add a sentence on ERVs and lncRNAs implicating a putative relationship as follows: in fact, ERVs themselves may be lncRNAs through transcription from their 3'LTR
Comments on the Quality of English Language
There are numerous minor grammatical errors and unnecessary capital letters in the manuscript, but I trust the text editor will do something about them
